# Polarization Anisotropies in Strain-Free, Asymmetric, and Symmetric Quantum Dots Grown by Droplet Epitaxy

**DOI:** 10.3390/nano11020443

**Published:** 2021-02-10

**Authors:** Marco Abbarchi, Takaaki Mano, Takashi Kuroda, Akihiro Ohtake, Kazuaki Sakoda

**Affiliations:** 1Aix Marseille Univ, Université de Toulon, CNRS, IM2NP Marseille, France; 2Research Center for Functional Materials, National Institute for Materials Science, 1-1 Namiki, Tsukuba, Ibaraki 305-0044, Japan; MANO.Takaaki@nims.go.jp (T.M.); KURODA.Takashi@nims.go.jp (T.K.); OHTAKE.Akihiro@nims.go.jp (A.O.); SAKODA.Kazuaki@nims.go.jp (K.S.)

**Keywords:** III–V quantum dots, droplet epitaxy, exciton dynamics

## Abstract

We provide an extensive and systematic investigation of exciton dynamics in droplet epitaxial quantum dots comparing the cases of (311)A, (001), and (111)A surfaces. Despite a similar s-shell exciton structure common to the three cases, the absence of a wetting layer for (311)A and (111)A samples leads to a larger carrier confinement compared to (001), where a wetting layer is present. This leads to a more pronounced dependence of the binding energies of s-shell excitons on the quantum dot size and to the strong anti-binding character of the positive-charged exciton for smaller quantum dots. In-plane geometrical anisotropies of (311)A and (001) quantum dots lead to a large electron-hole fine interaction (fine structure splitting (FSS) ∼100 μeV), whereas for the three-fold symmetric (111)A counterpart, this figure of merit is reduced by about one order of magnitude. In all these cases, we do not observe any size dependence of the fine structure splitting. Heavy-hole/light-hole mixing is present in all the studied cases, leading to a broad spread of linear polarization anisotropy (from 0 up to about 50%) irrespective of surface orientation (symmetry of the confinement), fine structure splitting, and nanostructure size. These results are important for the further development of ideal single and entangled photon sources based on semiconductor quantum dots.

## 1. Introduction

Droplet epitaxy [1,2,3,4] and droplet etching [5,6,7,8,9,10,11,12,13] (DE) are alternative growth protocols to Stranski-Krastanov for the fabrication of strain-free III–V-based semiconductor quantum dots (QDs). This emerging class of nanostructures has been efficiently exploited for the fabrication of classical optoelectronic devices such as photodetectors [14], lasers [15,16,17,18,19], and quantum emitters [13,20,21,22,23,24,25,26,27,28,29,30,31,32], demonstrating the relevance of this approach for realistic applications.

DE QDs are grown by: (1) formation of liquid Ga droplets by a supply of Ga (in absence of As in the molecular beam chamber), (2) followed by their crystallization into GaAs by a supply of As flux. Initial Ga droplets formed on the surfaces exhibit a perfect hemispherical shape [33,34]. As flux intensity and substrate temperature for crystallization are the key parameters determining the final shape of the nanostructures that can be tuned with unprecedented versatility [35]. Individual QDs; QDs diads [2,36,37]; single and multiple quantum rings (QRs) [16,21,38,39,40,41,42,43]; hybrid QD–QR structures, such as ring-on-a-disk [44], dot in-a-ring [45], or dot-on-a-disk [46]; and quantum wires [47] are remarkable examples of the possibilities offered by this self-assembly technique. In addition to this, DE allows forming QDs without a wetting layer [47,48,49,50,51,52,53,54,55], and their size and density can be independently tuned [56]. These features are not easily obtained with other methods such as conventional Stranski-Krastanov growth based on accumulation and relaxation of strain [57].

An appealing possibility offered by DE is the growth of nanostructures on different substrate orientations: in addition to the conventional (001), QDs can be grown on the three-fold symmetric (111)A surface [25,26,49,56,58,59,60,61,62,63,64,65] (e.g., for the fabrication of sources of entangled photons) and on the highly anisotropic (311)A surface [17,34,47,50,51,52,66,67,68,69] (e.g., to obtain large QDs density for laser emission). These substrate orientations affect the QDs properties and, in turn, their photophysics by modifying the confining potential (e.g., symmetry) for electrons and holes, and are thus a key tool for engineering the exciton dynamics and the corresponding optical properties.

Here, we provide a systematic comparison of the exciton dynamics of strain-free DE QDs grown on (311)A-, (001)-, and (111)A-oriented substrates studied using the photoluminescence (PL) spectroscopy of individual emitters. We show that the presence or absence of a wetting layer underneath the QDs modifies the confinement regime, providing a different evolution of the Coulomb interactions between electrons and holes. The presence of asymmetries in the excitonic confining potential lifts the degeneracy of the neutral exciton state, leading to the presence of a large fine structure splitting (FSS) for (311)A and (001) QDs, whereas this figure of merit is abruptly reduced for (111)A QDs. In all the samples, irrespective of surface orientation, the trend in the FSS does not show any significant size dependence. Similarly, the linear polarization anisotropy (the fingerprint of mixing between heavy-hole/light-hole states (*hh/lh*)) is randomly distributed and does not show any dependence on size or geometrical asymmetry.

## 2. Materials and Methods

### 2.1. Sample Fabrication

All the samples were grown on semi-insulating GaAs substrates by conventional solid-source molecular-beam epitaxy system (MBE, MBE32 by Riber, Bezons, France). The processes are described as follows.

For sample (311)A [69], firs,t a 2 μm thick Al${}_{0.55}$Ga${}_{0.45}$As layer was grown at 500 °C followed by a 136 nm thick Al${}_{0.26}$Ga${}_{0.74}$As core-layer grown at 610 °C. At the center of the core-layer, GaAs QDs were formed by droplet epitaxy: nominal 1.5 monolayers of Ga were grown at a speed of about 0.1 monolayers per second, supplied in the absence of As${}_{4}$ flux at 275 °C for droplets formation. The droplets were then crystallized into GaAs QDs by supplying a flux of As${}_{4}$ (2 × 10${}^{-6}$ Torr beam equivalent pressure) at 200 °C. The temperature was increased up to 400 °C for 10 min under As_4_ to improve the crystal quality of the QDs. The QDs were capped with a 30 nm thick Al${}_{0.26}$Ga${}_{0.74}$As at 400 °C, and the rest of the Al${}_{0.26}$Ga${}_{0.74}$As (38 nm) layer was grown at 625 °C. Finally, once the entire growth sequence was completed, a rapid thermal annealing process was performed at 785 °C for 4 min in an As_4_ atmosphere to improve the optical quality [16,70].

For sample (001) [71], first, a thick Al${}_{0.3}$Ga${}_{0.7}$As barrier layer was grown at 580 °C. Then, the substrate temperature was lowered to 350 °C together with a reduction in the As pressure. At this point, 1.5 monolayers of Ga were supplied for Ga droplet formation. Then, the As${}_{4}$ flux was increased to 2 × 10${}^{-4}$ Torr (beam equivalent pressure) to crystallize the Ga droplets into GaAs QDs at 200 °C, which were annealed in situ at 400 °C for 10 min under As${}_{4}$ flux. We then grew a 40 nm thick Al${}_{0.3}$Ga${}_{0.7}$As capping layer by standard molecular beam epitaxy (MBE) at 400 °C, followed by the growth of a 20 nm thick Al${}_{0.3}$Ga${}_{0.7}$As layer at 580 °C. A 10 nm thick GaAs capping layer was grown by standard MBE at at 580 °C. Finally, the sample was processed with post-growth annealing [16,70].

For sample (111), first, a thick Al${}_{0.3}$Ga${}_{0.7}$As barrier layer was grown at 500 °C. Then, the substrate temperature was lowered to 400 °C together with a reduction in the As pressure. At this point, 0.05 monolayers of Ga were supplied for Ga droplet formation. The As${}_{4}$ flux was set to 2 × 10${}^{-6}$ Torr beam equivalent pressure to crystallize the Ga droplets into GaAs QDs at 200 °C, which were then annealed in situ at 500 °C for 10 min under As${}_{4}$ flux irradiation. A 50 nm thick Al${}_{0.3}$Ga${}_{0.7}$As capping layer was grown by standard MBE at 500 °C followed by the growth of a 10 nm thick GaAs layer at 500 °C. Finally, the sample was annealed at 600 °C under As${}_{4}$ flux to improve the optical quality [16,70].

Cross-sectional TEM and scanning tunneling microscopy (STM) images, supported the high crystal quality [17,49,72] and no signatures of arsenic antisite defects, as could be, in principle, expected from low temperature growth, were observed. A shortening of the emission lifetime associated with this kind of defect was not observed as demonstrated by the systematic time-resolved measurements on ensemble and individual QDs [26,64,73].

The presence or absence of a wetting layer underneath DE QDs has been extensively studied [16,33,49,51,72]. Here, we summarize the main phenomenology for the three surfaces.

On the GaAs(100) surface, a wetting layer formed just before the droplet formation. Normally, we first supply Ga on the c(4 × 4) reconstructed surface where 1–1.75 monolayers of excess As atoms are present [74]. The first 1–1.75 monolayers of Ga atoms combine with these excess As, forming a GaAs wetting layer [75]. At this point, Ga droplets start forming. The wetting layer formation was clearly confirmed by cross-sectional observations [16,33,49,51,72].

On the (311)A ((8 × 1) reconstruction) and (111)A surfaces ((2 × 2) reconstruction), the surfaces are originally Ga-rich [76,77]. On these surfaces, droplet nucleation occurs immediately after the supply of Ga (even less less than one monolayer of Ga [49] and the formation of a two-dimensional GaAs wetting layer is suppressed. In principle, a wetting layer can be intentionally introduced on these surfaces by growing a thin GaAs layer on the AlGaAs barrier before the droplet [78,79].

### 2.2. Microscopy for Morphological Characterization

QDs morphology was studied on samples left uncapped. We used an atomic force microscope (AFM, SPA400 by Hitachi High-Tech, Tokyo, Japan) in non-contact mode and an in situ STM microscope (only for the (111)A sample).

### 2.3. Optical Spectroscopy

For photoluminescence spectroscopy measurements (PL) of individual nanostructures, the samples were kept in a liquid-helium cryostat at about 10 K. The PL signal was collected with a custom-made confocal spectroscopic setup with a diffraction-limited lateral resolution of about 1 μm. Excitation was performed above the barrier energy with a CW laser at 532 nm (about 2.3 eV). The PL signal was analyzed with a spectrometer and detected by a Si-based CCD camera, allowing for a spectral resolution better than 25 μeV in full width at half maximum (FWHM). The PL signal was discriminated in polarization using a linear polarizer and a half wave plate.

## 3. Results and Discussion

### 3.1. Morphology of Droplet Epitaxial Quantum Dots on (311)A, (001), and (111)A Surfaces

We first provide a description of the main morphological features of QDs grown on differently oriented surfaces. Although for the present work we selected three typical samples for extensive characterization, similar ones were implemented on the three surfaces while maintaining a comparably high quality (e.g., sharp line-width). More importantly, at least for some relevant cases, other groups have independently obtained very similar samples showcasing, for instance, quantum emission and entanglement from the (001) and (111)A surfaces with both droplet epitaxy and droplet etching [4,27,30,32].

The three samples ((311)A, (001), and (111)A)) were characterized by AFM (Figure 1). To ensure thoroughness, the QDs that were selected for AFM were likely much larger than those characterized in PL as suggested in previous papers: [80,81] provided a exciton Bohr radius of about 11 nm in GaAs; we expected to observe marked quantum confinement effects in much smaller QDs, as also suggested by the binding energy measurements (see below and [66]) and magneto-optical measurements [82]. The effect of capping after QDs formation should not affect the final shape of the nanostructures [16].

QDs grown on the (311)A surface have a rather complex morphology (Figure 1a), which is ascribed to the low AS${}_{4}$ pressure and the strong asymmetry of the underlying substrate [17,34,47,50,51,52,67,68]. Due to the non-equivalent adatom diffusion along the oppositely oriented directions [−233] and [2–3–3], the crystallization process from metallic droplets into GaAs QDs occurs at different rates on one side ([−233]). Instead of forming rings with a central hole for the conventional (001) case [16,21,38,39,40,41,42,43,69], they feature a U-shape with two protrusions along the [−233] direction, which is specific to this surface orientation.

QDs grown by DE on the conventional (001) surface have been extensively studied [71,83,84,85,86] (Figure 1b). The sample studied here was grown with an improved DE method, including a high temperature annealing step at 400 °C [71]. This step is crucial for improving the crystal quality (e.g., to reduce the formation of As precipitates in the top barrier [87]) and, in turn, the PL properties of the exciton recombination. A side effect of this annealing step is an increase in the elongation of QDs: typically, this kind of sample features a more pronounced asymmetry along the [1–10] direction (which is the direction of maximal surface diffusion with this surface orientation) with respect to samples grown at lower temperature [83] and lesser optical quality.

QDs grown on the (111)A surface are typically more symmetric compared to the previous two cases due to the three-fold symmetry of the underlying substrate orientation [25,26,49,56,58,59,60,61,62,63,64,65]. Triangular and hexagonal structures can be formed depending on the growth conditions [49,58]. In vacuo STM and high-resolution micrographs from a (111)A sample (grown in similar conditions to those studied in PL) revealed that the QDs are composed by piles of terraces, forming a truncated pyramid and do not feature any preferential elongation in a specific direction (Figure 2).

### 3.2. s-Shell Excitons in Droplet Epitaxial Quantum Dots

PL spectroscopy is systematically used to investigate the electronic properties of QDs. In the following sections, we address the main PL components of (311)A, (001), and (111)A QDs. As an example, we show the cases of (001) and (111)A QDs (Figure 1a,b),respectively). The other case, (311)A, is characterized by the same s-shell structure and is not explicitly reported. For a thorough investigation of this case, see reference [69].

For the conventional case of Stranski-Krastanov growth, s-shell excitons in DE QDs are the neutral exciton and biexciton (X and XX, respectively) and positively and negatively charged excitons (X${}^{+}$ and X${}^{-}$, respectively) (Figure 3). The assessment of each spectral component of the s-shell excitons has been largely discussed in previous works [26,49,66,69,71,83,88,89,90,91] and is not repeated here.

From this kind of spectrum, we can obtain various information concerning the exciton dynamics: (i) the binding energy of XX, X${}^{+}$, and X${}^{-}$, measured as the energy distance between an excitonic PL and the corresponding X line (Figure 3a top panel); (ii) the fine Coulomb interaction between carriers spins, measured as the energy splitting of the two linearly polarized components of X and XX PL lines (Figure 3a, bottom panel); and (iii) the mixing between *hh* and *lh* states, measured from the polar diagram of PL intensity as a function of the detected polarization angle (see below).

These three important features of the photophysics of excitons in QDs were comparatively addressed for the (001), (111)A, and (311)A cases. In the specific case under study and unlike the Stranski-Krastanov counterpart, these figures of merits are not influenced by strain and related piezoelectric fields, thus simplifying the overall picture and allowing directly linking the observation to the geometrical parameters of the nanostructures (e.g., asymmetry and orientation with respect to the crystallographic axes) and to heavy-hole, light-hole mixing.

### 3.3. Binding Energy of s-Shell Excitons

The emission energy of all the PL lines with respect to the corresponding X line can be used to directly assess the binding energy (BE): the energy difference between X and the other lines is a measure of the Coulomb interactions within an excitonic complex [66,92,93,94]. We performed this analysis for the three surfaces on a large number of QDs, spanning an interval of about 200 meV of X emission energy (Figure 4). The results for (001) and (311)A samples were reproduced from reference [66]. For thoroughness, we mention that the interval of X emission energy is slightly lower for the (311)A case due to a lower Al content in the barrier material in this sample with respect to the others.

The overall feature common to all the samples is described as follows [66]: XX and X${}^{-}$ have a binding character whereas X${}^{+}$ changes from binding to anti-binding when increasing X emission energy (that is, reducing QD size). More precisely, the BE of X${}^{-}$ shows a weak increase (although its behavior is scattered for (311)A and (001) cases, and this exciton complex is not observed in smaller QDs) whereas that of XX, X${}^{+}$ decreases. The BE of XX and X${}^{+}$ changes smoothly for the (001) case (changes are in the 4 meV range; Figure 4b), whereas the changes are more abrupt for the (311)A and (111)A cases (5–10 meV, Figure 4a,c).

This complex behavior is determined by the combination of attractive and repulsive Coulomb forces between the carriers composing the different excitonic species. Due to the presence of several particles composing the excitons in the s-shell, the overall picture can be qualitatively understood by considering the mean-field corrections and correlation effects necessary to account for the many-body characteristic of the problem. The larger *hh* mass with respect to the *e* one in GaAs results in a larger probability density, providing a stronger localization of the hole in the QD and a reduced penetration into the barrier material. Correspondingly, a larger carrier density is found for *hh* with respect to *e* [95]. This difference is responsible for a larger repulsive interaction in X${}^{+}$, which leads to decreasing the BE and an anti-binding character for smaller nanostructures. For the same reason, the interaction between an *e* and a *hh* results in a mean field that is attractive for an extra electron, leading to the binding character of X${}^{-}$ and to an increase in the BE of smaller QDs, as shown by the experimental data (Figure 4).

A more refined theoretical framework based on a quantum Monte-Carlo approach for strain-free QDs [96,97,98]was developed [66] to account for the difference between a truly 3D confinement compared to a weaker lateral confinement. This model explains the differences between the steep changes in the BE for excitons in (311)A and (111)A cases with respect to the (001) case. In the former scenario, the lack of wetting layer in (311)A and (111)A results in a stronger carrier confinement and thus in an overall picture that can be captured by the spherical potential (strong confinement in all the directions). In the case of (001), the presence of a wetting layer results in a weaker lateral confinement (strong confinement only along the vertical growth direction) and a behavior that can be described by a shallow disk. Thus, the presented results confirm that in (111)A QDs, the same strong confinement regime holds as for the (311)A case [66].

Although a direct assessment of the QDs size is impossible, we can roughly estimate the size changes based on the agreement between the data and the theoretical model shown in reference [66] as well as on magneto-optical measurements [82]. We deduced that the lateral size of the QDs for the (311)A and (111)A cases changes from about 12 nm in diameter for low-energy QDs up to a minimum of about 7 nm for the high-energy ones. For (001) QDs, we estimated a change from about 10 to 4 nm.

### 3.4. Electron-Hole Spin Interactions, Fine Structure Splitting

The study of the linear polarization dependence of the s-shell excitons emission allows measuring the polarization splitting of each PL line (Figure 3b) [99]. Generally, X and XX show mirror symmetric polarization splitting, which is the fingerprint of fine electron-hole interactions (fine structure splitting) [83,100,101,102]. X${}^{+}$ does not show any energy splitting, with the fine structure splitting occurring in the initial state (two holes in the valence band with opposite spin interacting with an electron in the conduction band) and in the final state (one hole in the valence band) of the recombination zero paths. A similar consideration holds for the negative-charged exciton counterpart X${}^{-}$. Beyond fundamental physics, this feature is extremely relevant for the implementation of entangled photon emission springing from XX–X-cascaded photon pairs [30,31,32,103], provided that the fine structure splitting (FSS) is negligible and the which-path-information is erased.

Here, we studied the FSS for three samples, exploring the cases of large anisotropy (311)A, lower anisotropy (001), and vanishing anisotropy (111)A (Figure 5). In all cases, no size dependence was observed with fluctuations in the 300 μeV range for (311)A and (001) cases and about one order of magnitude lower for the (111)A case (Figure 5a–c). The statistical analysis showed an average FSS of 155 ± 66 μ eV for (311)A, 134 ± 62 μ eV for (001) and 16 ± 9 μ eV for (111)A, the error being the standard deviation of the distribution (Figure 5d–f).

These results confirmed that the anisotropic surface diffusion of GaAs adatoms during QDs growth atop AlGaAs results in large shape anisotropies that break the in-plane symmetry of the QDs. Despite the lack of piezoelectricity in these strain-free nanostructures, the geometric anisotropy promotes the fine interaction between the electron and hole spins in the X state, lifts its degeneracy, and leads to energy distinguishable recombination paths from the XX to the X state, limiting the possibility of obtaining entangled photon pairs. Thus, although the FSS can be reduced a posteriori via strain tuning [86], extending the use of anisotropic surfaces for the fabrication of entangled photon sources, the use of three-fold symmetric (111)A surfaces is clearly favorable for this task [26,27,29]. Alternatively, droplet etching can produce extremely small fine structure splitting using a (001) surface [10].

From the same investigation, we determined the directions of the polarization axes of the split PL components with respect to the crystallographic axes (Figure 3b and Figure 6). As previously reported for the (001) case [104], for high quality DE nanostructures grown at higher temperature, the morphology of the QDs showed a marked elongation along the [1–10] in-plane direction (that is, as the basis of the FSS measured in these kind of samples) and the two orthogonally polarized PL components were mostly aligned along these axes. This observation holds for the large majority of the QDs grown on the (311)A and (001) surfaces but it was not verified in the (111)A case (Figure 6a,b). By reporting the angle of the high-energy split component ($\theta (E_{max}\left(X\right))$) for the (111)A case as a function of the corresponding X emission energy and the corresponding statistical distribution (Figure 6a,b respectively), we observed a completely random behavior. Information obtained from in vacuo STM imaging (Figure 2) justified the lack of dependence of the FSS on QDs size or a preferential axis for the two PL split components with their shape not oriented along a particular axis.

A similar investigation performed for (311)A QDs showed that preferential directions of the polarization axes appeared as expected for asymmetric structures (Figure 6c,d). The case of (001) is not reported explicitly; however, with few exceptions, all the QDs on this surface are well-oriented along the [110] in plane direction, as already shown in previous works on this sample [66,104].

### 3.5. Heavy-Hole, Light-Hole Mixing

Another feature emerging from the characterization of the PL spectra of single QDs when detected as a function of the polarization angle is a differing intensity of the two split components of X and XX (Figure 3b and Figure 7). This is the signature of the relevant mixing between *hh* and *lh* states, leading to non-pure selection rules for electron and hole recombination [105,106,107,108,109,110,111,112]. In III–V QDs, the bottom of the conduction band can be described, in a first approximation, with an isotropic and parabolic dispersion. Valence bands, instead, are characterized by the presence of *hh* and *lh* states, which are tens of meV apart. Their bands dispersion features a large anisotropy and different curvatures (different effective masses). In Stranski-Krastanov QDs the presence of piezoelectricity results in a large *hh–lh* mixing [106,107,109,110,111] and, finally, uneven PL intensity for the recombination paths of electrons and holes owing to different selection rules.

The same features were first found in (001) strain-free DE QDs with shape asymmetries [105,108]. The presence of elongation along the crystallographic axis results in the uneven PL intensity of the two orthogonally polarized components of the X line. This feature can be conveniently represented in a polar plot of the high- and low-energy components of the X split doublet (Figure 7a, top panel). The same effect can be observed in all the s-shell excitons (Figure 3b), allowing us to focus, for instance, on the X${}^{+}$ line only (Figure 7a,b, bottom panels). This possibility is particularly convenient for QDs with a very small FSS, such as those grown on the (111)A substrate, where the two PL lines of the X split doublet cannot be easily resolved due to the limited spectral resolution of the spectrometer in use and the comparably large FSS and line broadening.

Another important feature to consider when explaining the polarization anisotropy and the effect of *hh–lh* mixing is the orientation of the QD in-plane elongation with respect to the crystallographic axes. An orientation of the polarization axes parallel to the in-plane, the main crystallography directions, is the signature of a geometrical elongation of the QD shape along these directions (as shown in Figure 7a) [105,107,108]. A different orientation of the geometrical elongation results in a rotation of the polar diagram with respect to the main crystallographic axes and in non-orthogonal relative orientation between the two PL split lines (as shown in Figure 7b) [105,107,108]. Note that the direction defined by the polarization diagram is not necessarily aligned to the in-plane elongation axis of the QD.

For the (311)A case, more than half of the QDs show an alignment along [01–1] or [−233] directions, whereas about 40% are randomly oriented (as shown in Figure 6c,d). Thus, despite the large anisotropy of this surface, the complex shape of the QDs can induce a disordered polarization orientation. This may be linked to small differences in the two protrusions of the QD along the [−233] direction (Figure 1a). For QDs grown on the (001) surface, we observed that the large majority showed an almost perfect alignment (within a few degrees) of the polarization axes along the [1–10] main crystallographic direction, with very few exceptions (such as that one shown in Figure 7b), accounting for their preferential geometrical elongation [105,107,108] in these directions (Figure 1b). Finally, for (111)A QDs, the lack of a preferential elongation of the QDs results in a completely randomized orientation of the polarization directions (see, for instance, Figure 6a,b).

Thus, given the larger symmetry of the (111)A QDs with respect to the others presented here, the corresponding FSS is smaller and the orientation of the polarization axes (and the underling geometrical anisotropy) is completely random. However, despite this larger symmetry, the polarization anisotropy ($\rho =\left(I_{max}-I_{min}\right)/\left(I_{max}+I_{min}\right)$) fluctuates in the same range for all threes cases. By plotting ρ as a function of the corresponding FSS (Figure 8) or X emission energy (not shown), we observed no dependence of the *hh–lh* mixing on shape anisotropy (FSS) or size (X emission energy) and no reduction in the (111)A case with respect to the others.

## 4. Conclusions

In conclusion, we showed that DE is a versatile technique for the growth of high quality III–V QDs on differently oriented substrates. The s-shell exciton dynamics are strongly influenced by the presence or absence of a wetting layer. A strong lateral confinement was found for (311)A and (111)A QDs in contrast with the (001) case, where the carrier confinement is weaker. This results in a marked difference in the Coulomb interaction between electrons and holes composing the s-shell excitons. The presence of geometrical asymmetries in the QDs shape leads to a rather large fine interaction between electrons and holes spins for the anisotropic surfaces of (311)A and (001), whereas for the three-fold symmetric (111)A case, this figure of merit is in the 10μ eV range. However, despite the increased symmetry of the confining potential, a relevant polarization anisotropy was also found for this latter case, pointing to a relevant mixing between heavy- and light-hole states in the valence band.

## Figures and Tables

**Figure 1 nanomaterials-11-00443-f001:**
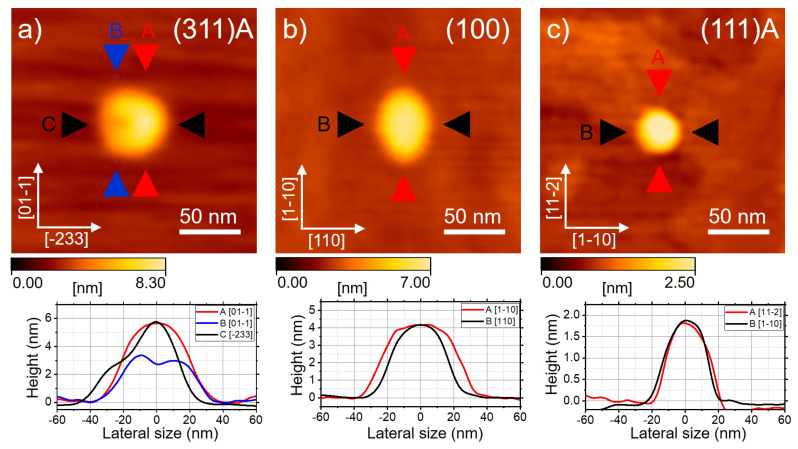
(**a**) Atomic force microscopy (AFM) image of a quantum dot (QD) grown on the (311)A surface for the sample described in [69]. (**b**) AFM image of a QD on the (001) surface for the sample described in reference [71]. (**c**) AFM image of a QD on the (111)A surface. In the bottom panels, we plot the cross-sectional profiles of the QDs in different crystallographic directions as marked in the images. The sizes of the QDs on the different substrates are not linked to the surface on which they were grown. The chosen QDs are representative of their kind regarding shape. However, they are likely incompatible with the photoluminescence (PL) spectra shown in the following figures due to their large size.

**Figure 2 nanomaterials-11-00443-f002:**
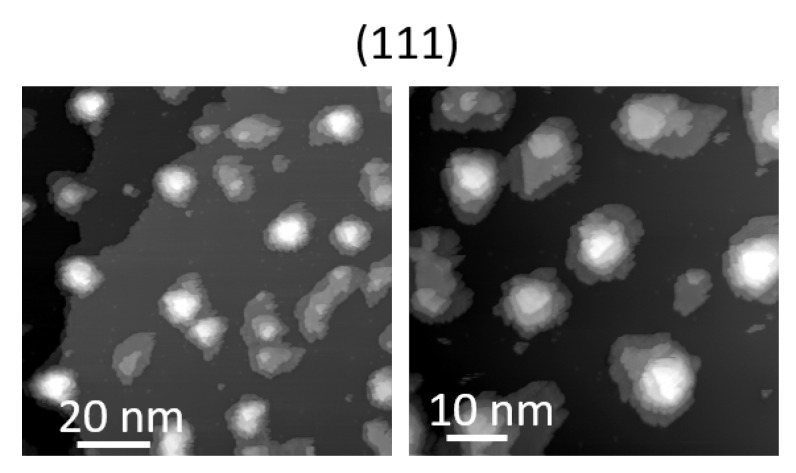
High-resolution in vacuo scanning tunneling microscopy STM micrographs of an ensemble of GaAs QDs sitting on (111)A-oriented AlGaAs substrate.

**Figure 3 nanomaterials-11-00443-f003:**
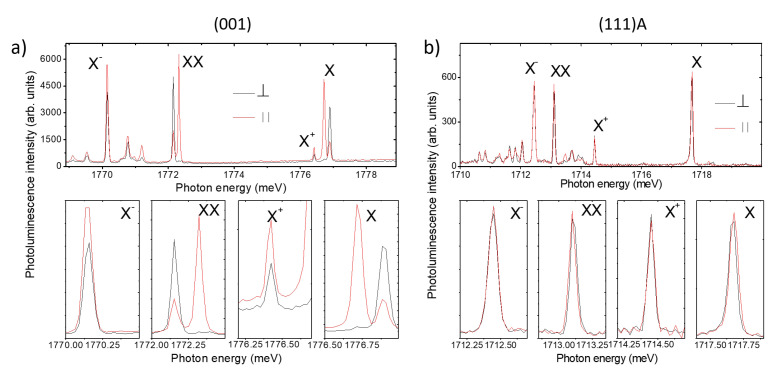
(**a**) Top panel: photoluminescence (PL) spectrum of a QD grown on the (001) surface for the sample described in reference [71]. The main excitonic recombination lines from the s-shell are highlighted. Red and black lines represent the orthogonally and linearly polarized components of the PL, respectively, oriented along the crystallographic axes [−110] and [1–10]. Bottom panels: From the left to the right panel are magnifications of each individual line in the spectrum: X${}^{-}$, XX, X, and X${}^{+}$ for both linearly polarized components, respectively. (**b**) The same as for (a) but for a QD grown on the (111)A-oriented substrate.

**Figure 4 nanomaterials-11-00443-f004:**
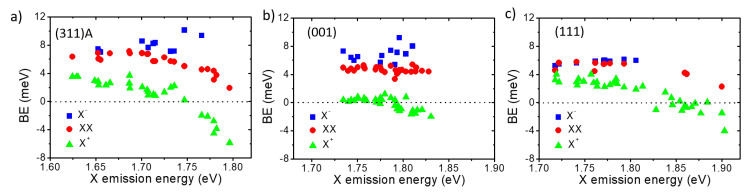
(**a**) Binding energy of XX, X${}^{+}$, and X${}^{-}$ for (311)A QDs as a function of X emission energy, re-plotted from the reference [66] (311)A surface for the sample described in [69]. (**b**) Same as for (a) but for (001) QDs, re-plotted from reference [66] for the sample described in reference [71]. (**c**) Same as for (a) but for (111)A QDs.

**Figure 5 nanomaterials-11-00443-f005:**
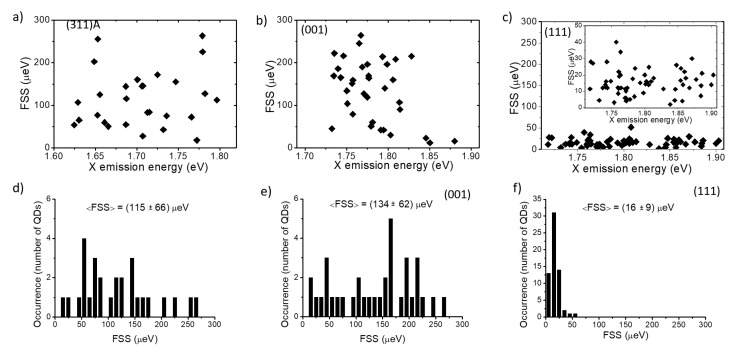
(**a**) Fine structure splitting (FSS) as a function of the X emission energy for (311)A QDs for the sample described in [69]. (**b**) Fine structure splitting as a function of the X emission energy for (001) QDs for the sample described in reference [71]. (**c**) Fine structure splitting as a function of the X emission energy for (111)A QDs. The insets show the same data rescaled for clarity. (**d**–**f**) Statistical distribution of the FSS corresponding to (**a**–**c**), respectively.

**Figure 6 nanomaterials-11-00443-f006:**
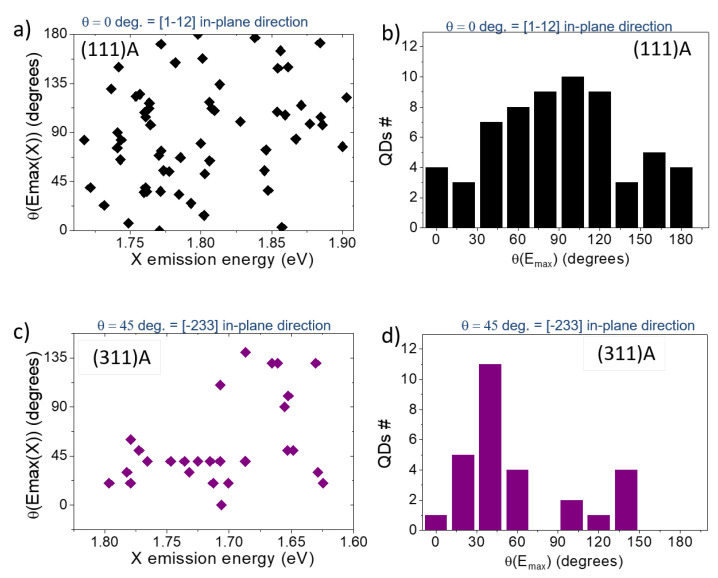
(**a**) Polarization angle $\theta (E_{max}\left(X\right))$ corresponding to the high energy split component of the X doublet for (111)A QDs as a function of the corresponding X emission energy obtained from a set of about 60 QDs. An angle of 0 degrees corresponds to the in-plane [1–12] direction. (**b**) Statistical distribution $\theta (E_{max}\left(X\right))$ for the (111)A QDs. (**c**) Polarization angle $\theta (E_{max}\left(X\right))$ corresponding to the high-energy split component of the X doublet for (311)A QDs as a function of the corresponding X emission energy obtained from a set of about 30 QDs. An angle of 45 degrees corresponds to the in-plane [−233] direction. (**d**) Statistical distribution of $\theta (E_{max}\left(X\right))$ for the (311)A QDs.

**Figure 7 nanomaterials-11-00443-f007:**
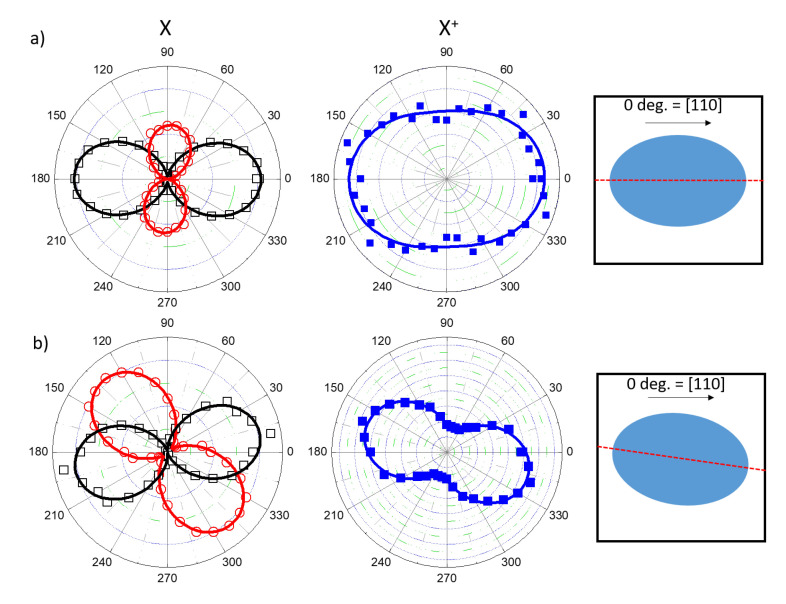
(**a**) Polar diagram of X PL intensity (left panel) and X${}^{+}$ PL intensity (central panel) for a QD grown on the (001) surface for the sample described in reference [71] and having a geometrical asymmetry oriented along the main crystallographic axes, as described in the scheme reported in the right panel. (**b**) Polar diagram of X PL intensity (left panel) and X${}^{+}$ PL intensity (central panel) for a QD grown on the (001) for the sample described in reference [71] surface and having a geometrical asymmetry axis tilted with respect with the main crystallographic axes, as described in the scheme reported in the right panel.

**Figure 8 nanomaterials-11-00443-f008:**
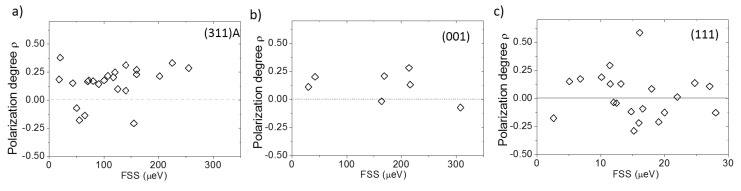
(**a**) Polarization degree of the X PL line as a function of the corresponding FSS for QDs grown on the (311)A surface for the sample described in [69]. (**b**) Same as for (a) but for (001) QDs for the sample described in reference [71]. (**c**) Same as for (a) but for (111)A QDs.

## Data Availability

The data presented in this study are available on request from the corresponding author.

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
