# Peer review of "Polarization Anisotropies in Strain-Free, Asymmetric, and Symmetric Quantum Dots Grown by Droplet Epitaxy"

_nanomaterials, 2021, doi:10.3390/nano11020443_

Round 1
Reviewer 1 Report
This paper systematically compared the micro-PL spectrum of three growth conditions in terms of binding energy, fine structure splitting, and polarization anisotropy over a wide spectral range. I think this work would be quite useful to the relevant researchers interested in droplet QDs. I agree most of the conclusions but the following minor revision is expected for readers.
(1) the PL spectrum of sample (111)A is necessary in comparison with the other two cases regarding its extremely small FSS. Perhaps in Fig 2
(2) In Fig 5, specifying the theta =0 would be helpful.
(3) According to Fig. 1, one may think the dot size becomes decreased while the growth condition changes from (311)A to (111) via (001). Apart from the main conclusion of this paper, micro-PL spectra are seen over a wide spectral range. Authors may explain the wide spectral distribution. The size distribution is not enough to explain this due to such a large diameter.
Author Response
REVIEWER 1
This paper systematically compared the micro-PL spectrum of three growth conditions in terms of binding energy, fine structure splitting, and polarization anisotropy over a wide spectral range. I think this work would be quite useful to the relevant researchers interested in droplet QDs. I agree most of the conclusions but the following minor revision is expected for readers.
(1) the PL spectrum of sample (111)A is necessary in comparison with the other two cases regarding its extremely small FSS. Perhaps in Fig 2 .
OK. We modified Figure 2 by adding the important case of 111A substrate by explicitly showing a detailed spectrum. The polar diagram is removed.
(2) In Fig 5, specifying the theta =0 would be helpful.
OK. We directly wrote on graphs a) and b) what was only specified in the corresponding caption:
“q = 0 deg. = [1-12] in-plane direction.”
(3) According to Fig. 1, one may think the dot size becomes decreased while the growth condition changes from (311)A to (111) via (001). Apart from the main conclusion of this paper, micro-PL spectra are seen over a wide spectral range. Authors may explain the wide spectral distribution. The size distribution is not enough to explain this due to such a large diameter.
OK. In order to clarify this point we added the following sentence in the caption:
“The sizes of the QDs on the different substrates are not linked to the surface they are grown on. The chosen QDs are representative of their kind for what concerns the shape. However, they likely are not compatible with the PL spectra show in the following figures owing to their large size.”
and later in the text:
“For the sake of thoroughness, we mention that the QDs that are selected for AFM are likely much larger than those characterized in PL as suggested in these papers [Mlinar2009,Luo2011]: provided a exciton Bohr radius of about 11 nm in GaAs, we expect to observe marked quantum confinement effects in much smaller QDs as also suggested by binding energy measurements (see later and reference ABBARCHI2010) and magneto-optical measurements [Abbarchi2010c].”
Reviewer 2 Report
Report on “Polarization anisotropies in strain-free, asymmetric and symmetric quantum dots grown by droplet Epitaxy” by Marco Abbarchi et al.
The manuscript compares the optical emission from GaAs QDs fabricated on substrates with different surface orientations. The topic is of interest, the paper is clearly written, the experiments are well described, and the analysis is mostly sound. Therefore, I recommend publication after minor revision of the points given below.
1) page1, line 17: Regarding the references related to droplet etching the author cites only papers from one research group ignoring, e.g., the first observation of this technique by Wang and also other authors in this field.
2) page 3, line 91: The obtained results are based on measurements from three samples. The authors should comment the reproducibility of the fabrication process.
3) page 6, line 155: The authors should explain what a “larger localization within the QD” means. Furthermore, a “carrier density” usually describes the density of charge carriers, e.g., in a doped semiconductor. Here they probably mean the probability density as given by the square of the wave function. The stronger localization of the hole probability-density in the QD can be explained by a reduced penetration into the barrier material.
4) page 6, line 160: In the discussion of the influence of a possible wetting layer, it is not clear which results are already known and which are new.
5) page 6, line 173: In the analysis of the polarization dependent experiments the authors seem to distinguish between “electron-hole fine interactions” and the “fine structure splitting (FSS)”. If yes, they should clarify the difference.
6) page 7, line 193: As a further alternative, GaAs QDs fabricated by droplet etching on (001) demonstrate a FSS of 3.9 µeV [Huo et al., Appl. Phys. Lett. 102 (2013)].
7) page8, fig.6b: The meaning of “but for a QD with an elongation tilted with respect to the main crystallographic axes.” is not clear. Is this QD also on (001)?
8) References: The authors give a high number of self-citations. They should check which of their cited papers are relevant for the present work.
Author Response
REVIEWER 2
Report on “Polarization anisotropies in strain-free, asymmetric and symmetric quantum dots grown by droplet Epitaxy” by Marco Abbarchi et al.
The manuscript compares the optical emission from GaAs QDs fabricated on substrates with different surface orientations. The topic is of interest, the paper is clearly written, the experiments are well described, and the analysis is mostly sound. Therefore, I recommend publication after minor revision of the points given below.
1) page1, line 17: Regarding the references related to droplet etching the author cites only papers from one research group ignoring, e.g., the first observation of this technique by Wang and also other authors in this field.
OK. We added the following references:
-Wang, Zhiming M., et al. "Self-organization of quantum-dot pairs by high-temperature droplet epitaxy." Nanoscale Research Letters 1.1 (2006): 57-61.
-Wang, Zh M., et al. "Nanoholes fabricated by self-assembled gallium nanodrill on GaAs (100)." Applied physics letters 90.11 (2007): 113120.
-Heyn, Ch, A. Stemmann, and W. Hansen. "Dynamics of self-assembled droplet etching." Applied Physics Letters 95.17 (2009): 173110.
-Heyn, Christian, et al. "Dynamics of mass transport during nanohole drilling by local droplet etching." Nanoscale research letters 10.1 (2015): 1-9.
-Heyn, Christian. "Kinetic model of local droplet etching." Physical Review B 83.16 (2011): 165302
-Stemmann, A., et al. "Local droplet etching of nanoholes and rings on GaAs and AlGaAs surfaces." Applied Physics Letters 93.12 (2008): 123108.
-Fuster, David, Yolanda González, and Luisa González. "Fundamental role of arsenic flux in nanohole formation by Ga droplet etching on GaAs (001)." Nanoscale research letters 9.1 (2014): 1-6.
2) page 3, line 91: The obtained results are based on measurements from three samples. The authors should comment the reproducibility of the fabrication process.
OK. We added the following sentence:
“Although for the present work we selected three typical samples for extensive characterization, similar ones have been implemented on the three surfaces while keeping a comparable high quality (e.g. sharp line-width). More importantly, at least for some relevant case, other groups have independently obtained very similar samples showcasing, for instance, quantum emission and entanglement from (001) and (111)A surfaces with both droplet epitaxy and droplet etching [Gurioli2019,Basso2018,Basset2019,Basset2020b}.”
3) page 6, line 155: The authors should explain what a “larger localization within the QD” means. Furthermore, a “carrier density” usually describes the density of charge carriers, e.g., in a doped semiconductor. Here they probably mean the probability density as given by the square of the wave function. The stronger localization of the hole probability-density in the QD can be explained by a reduced penetration into the barrier material.
Yes. The referee is right. We rephrased the sentence:
“The larger hh mass with respect to the e one in GaAs results in a larger probability density providing a stronger localization of the hole in the QD and a reduced penetration into the barrier material. Correspondingly, a larger carrier density is found for hh with respect to e [lelong1996].”
4) page 6, line 160: In the discussion of the influence of a possible wetting layer, it is not clear which results are already known and which are new.
OK. We added the following sentence:
“Results for (001) and (311)A samples are reproduced from reference [ABBARCHI2010].”
5) page 6, line 173: In the analysis of the polarization dependent experiments the authors seem to distinguish between “electron-hole fine interactions” and the “fine structure splitting (FSS)”. If yes, they should clarify the difference.
Actually, no. There is no difference. In order to clarify we modified the text as follows:
“[…] being the fingerprint of electron-hole fine interactions (fine structure splitting).”
In the rest of the paper we changed “fine interactions” with “fine structure splitting” to avoid confusion.
6) page 7, line 193: As a further alternative, GaAs QDs fabricated by droplet etching on (001) demonstrate a FSS of 3.9 µeV [Huo et al., Appl. Phys. Lett. 102 (2013)].
OK. We added the following sentence:
“Alternatively, also droplet etching can bring extremely small fine structure splitting using a (001) surface [Huo2013].”
7) page8, fig.6b: The meaning of “but for a QD with an elongation tilted with respect to the main crystallographic axes.” is not clear. Is this QD also on (001)?
Yes. It is a (001) QD. We rephrased the caption of Figure 6 b that now reads as follows:
“Polar diagram of X PL intensity (top panel) and X+PL intensity for a QD grown on the (001) surface and having a geometrical asymmetry axis tilted with respect with the main crystallographic axes.”
8) References: The authors give a high number of self-citations. They should check which of their cited papers are relevant for the present work.
As also acknowledged by reviewer 4 the authors are pioneers of droplet epitaxy and contributed to all major achievements in the filed, including growth, spectroscopy and device fabrication along more that 20 years. It is thus normal that in a comparative study as this work is, many citations point to their own work. We thus choose to keep the citations.
%%%%%%%%%%%%%%%%%%%%%%%%%%%%%%%%%%%%%%%%%%%%%%%%%%%%%%%%%%%%%%%%%%%%%%%%%%%%%%%%%%%%%%%%%%%%%%%%%%
Reviewer 3 Report
- Methods section. Description of QDs formation by droplet epitaxy. What is the key factor determining the shape of QDs on different crystalline orientation of substrates, as well different atoms on the substrate surface (Ga or As in 311, 100 and 111). Authors could explain the starting condition of growth GaAs QDs. What is the shape of droplets before the growth of GaAs QDs? This U shape is determined by Ga droplet shape?
- Since authors form QDs at low temperatures and after epitaxy improve the crystalline quality by annealing at higher temperature for several minutes, the probability of formation of arsenic antisites defects are high. Did authors studied the influence of antisites to PL properties?
- How these defects if appear could influence the anisotropy of polarization features?
Author Response
REVIEWER 3
Methods section. Description of QDs formation by droplet epitaxy. What is the key factor determining the shape of QDs on different crystalline orientation of substrates, as well different atoms on the substrate surface (Ga or As in 311, 100 and 111). Authors could explain the starting condition of growth GaAs QDs. What is the shape of droplets before the growth of GaAs QDs? This U shape is determined by Ga droplet shape?
The shape of the Ga droplets on these various surfaces is perfect hemispherical shape since they are liquid. Therefore, the droplet shape is not the key factor determining the shape of QDs. During the crystallization of these droplets by supplying As4 flux, the shape is determined by how the droplets crystallize into GaAs. As reported previously, there are many parameters determining the shapes such as surface orientation, As flux intensity, temperature, original size of the droplets, and surface anisotropy of the surfaces. To explain more about these, the following sentences are added.
“DE QDs are grown by: (1) formation of liquid Ga droplets by a supply of Ga (in absence of As in the molecular beam chamber), followed by (2) their crystallization into GaAs by a supply of As flux. Initial Ga droplets formed on the surfaces exhibit perfect hemispherical shape [33,34]. As flux intensity and substrate temperature for crystallization are the key parameters determining the final shape of the nanostructures that can be tuned with unprecedented versatility[35]. Individual QDs (QDs), Qds diads [2,36,37], single and multiple quantum rings (QRs)[16,21,38–43], hybrid QD-QR structures such as ring-on-a-disk[44], dot in-a-ring[45] or dot-on-a-disk[46], and quantum wires[47] are remarkable examples of the possibilities offered by this self-assembly technique. In addition to this, DE allows to form QDs without a wetting layer [47–55] and tune independently their size and density [56]. These features are not easily obtained with other methods such as conventional Stranski-Krastanov growth based on accumulation and relaxation of strain [57].
Since authors form QDs at low temperatures and after epitaxy improve the crystalline quality by annealing at higher temperature for several minutes, the probability of formation of arsenic antisites defects are high. Did authors studied the influence of antisites to PL properties?
In order to address this point we added the following sentence in the description of the QDs growth:
“Cross-sectional TEM and STM images, account for high crystal quality [17,49,72] and no signatures81of arsenic antisite defects, as could be in principle expected from lo temperature growth. In fact, a82shortening of the emission lifetime associated to such kind of defects is not observed as accounted by83systematic time resolved measurements on ensemble and individual QDs [26,64,73].”
How these defects if appear could influence the anisotropy of polarization features?
Based on the previous discussion we do not expect any influence of these kind of defects on the optical polarization
Reviewer 4 Report
The peer-reviewed manuscript by Abbarchi et al. continues a series of works by this group on the study of fundamental optical properties of GaAs QDs of various shapes, sizes, symmetries and surroundings, which are grown using droplet epitaxy (DE) on GaAs substrates with various orientations. The authors have a rich experience in this field since the 2000s and have published numerous articles detailing the different types of QDs. First, this manuscript is in many ways the first attempt by the authors to review and systematize the results obtained by this and other groups for the study of various QDs grown with the DE. In a recent manuscript in Nanomaterials (2020), 10, 1833, the authors have described in detail the asymmetric QDs grown on a (311)A oriented surface, and in the reviewed paper they thoroughly compare these results with the QDs with a similar (or even stronger) geometric asymmetry due to using for QD growth of an anisotropic substrate with (001) orientation. As a counterpart for these asymmetric samples the authors use three-fold symmetrical QDs grown on (111)A surface. In addition, the effect of wetting layers, occurring only in the QDs of the latter type, on the charge carrier confinement and the binding energies of the excitons was studied.
The main advantage of this manuscript is a clear demonstration of a difference in the exciton binding energy and the characteristics of the fine structure splitting for the two types of the QDs with different symmetry, as well as the revealing of the influence of wetting layers on the QD’s properties. The authors have studied these tasks intensively and widely used their own previous and literature data here. This part of the manuscript has rather review character. At the same time, the study of polarization characteristics of the QDs of various type is the most original part of manuscript.
Thus, I think this manuscript will be of interest to a wide range of physicists and technologists working in the field of optical studies of quantum-sized heterostructures and their fabrications by different epitaxial techniques. However, I would like to ask about several places that were not clear to me and to suggest some corrections and addings.
- In Section 2
1.1. The submitted text about sample preparation looks like a preliminary version of this part. Therefore, it must be rewritten in accordance with the common rules for describing the sample fabrication (as it was perfectly described in your previous manuscripts). Some details of specific recipes can be found in supplementary, which can be added to the manuscript.
1.2. The different samples of QDs grown on (001) surface are used in the different figures of the manuscript: one is described in Fig. 2 and Fig.6, while another ones appear in Fig.3c (reference to [38]) and fig.4b (without reference). The reference to only old reference [46] should be corrected and the equality or differences of the growth processes should be compared.
1.3. Section 2.2 describing the microscopes should be added with renumbering of optical spectroscopy to 2.3.
- In Section 3
2.1. The A, B, and C highlight cuts in different crystallographic directions would be very helpful in Figure 1a,b,c. (as it has been added in your previous publication in Nanomaterials). They can be added as insets to the figures with simultaneous displacement of the QD images.
2.2 In section 3.1 a brief discussion about wetting layers will be desirable to illustrate formation/absence of this layers by TEM images. What are the main factors determining the arising of wetting layers in the QD-structures grown on the various substrates used in this work? In the previous work published in Nanomaterials, 2020, 10, 1833, the possibility of growth QDs on the surface (311)A with and without wetting layer has been reported. Did you measure binding energies and FSSs for these structures?
2.3. I recommend transferring STM images from Fig.5c to Section 3.1 and to describe them here.
2.4. I recommend adding the title “Photon energy (meV)” for the bottom axis in fig. 2b. In addition, it would be desirable to add the bar illustrating the PL intensity plotted in this figure. Please also check the intensities of the linearly polarized components in Fig. 2a and 2b (if I understood correctly the brightness of the strokes in Fig.2b as intensity).
2.5. All axes in Fig.3 should be of the same thickness (slightly thicker than those used) and the symbols should be drawn with thicker lines (or filled as in Fig.4).
2.6. Line 157,159. Please explain what «smaller nanostructures» are? Moreover, abstract reports about size dependences. However, neither absolute nor comparative analysis of QD sizes has been performed in the manuscript. Thus, information on QD’s size distribution must be added in Section 3.1.!
2.7. All axes in Fig.4 should be of the same thickness (slightly thicker than those used).
2.8. In the caption in Fig.5 it would be desirable to add “…for an array of different (111)A QDs…” to emphasized the fact that the PL measurements were carried out for an array of the QDs. Moreover, the following text about “…the polarization analysis shown in Figure 2b” should be deleted or modified because this figure shows this analysis for QDs grown on the (001) surface.
2.9. In addition, the free space in figure 5 without STM images (see p.2.2) can be used to demonstrate a similar dependence of the polarization angle on the PL peak energy measured for one of the types of asymmetric QDs grown on the (001) or (311)A surfaces. A comparative analysis of these dependences for symmetric and asymmetric QDs should be added to the text to this figure.
2.10. A legend describing the orientation of the linearly polarized PL components denoted by the red circles and black squares should be added in Figure 6. There may be a schematic sketch illustrating the orientation of the polarization and crystallographic axes, as well as the crystallographic direction of the QD elongation will be helpful for the reader in this figure.
2.11. Lines 237-246. The discussion on the figure proposed in p.2.9 will be helpful in this paragraph.
Thus, the submitted manuscript can be published after moderate revision.
Author Response
%%%%%%%%%%%%%%%%%%%%%%%%%%%%%%%%%%%%%%%%%%%%%%%%%%%%%%%%%%%%%%%%%%%%%%%%%%%%%%%%%%%%%%%%%%%%%%%%%%
REVIEWER 4
The peer-reviewed manuscript by Abbarchi et al. continues a series of works by this group on the study of fundamental optical properties of GaAs QDs of various shapes, sizes, symmetries and surroundings, which are grown using droplet epitaxy (DE) on GaAs substrates with various orientations. The authors have a rich experience in this field since the 2000s and have published numerous articles detailing the different types of QDs. First, this manuscript is in many ways the first attempt by the authors to review and systematize the results obtained by this and other groups for the study of various QDs grown with the DE. In a recent manuscript in Nanomaterials (2020), 10, 1833, the authors have described in detail the asymmetric QDs grown on a (311)A oriented surface, and in the reviewed paper they thoroughly compare these results with the QDs with a similar (or even stronger) geometric asymmetry due to using for QD growth of an anisotropic substrate with (001) orientation. As a counterpart for these asymmetric samples the authors use three-fold symmetrical QDs grown on (111)A surface. In addition, the effect of wetting layers, occurring only in the QDs of the latter type, on the charge carrier confinement and the binding energies of the excitons was studied.
The main advantage of this manuscript is a clear demonstration of a difference in the exciton binding energy and the characteristics of the fine structure splitting for the two types of the QDs with different symmetry, as well as the revealing of the influence of wetting layers on the QD’s properties. The authors have studied these tasks intensively and widely used their own previous and literature data here. This part of the manuscript has rather review character. At the same time, the study of polarization characteristics of the QDs of various type is the most original part of manuscript.
Thus, I think this manuscript will be of interest to a wide range of physicists and technologists working in the field of optical studies of quantum-sized heterostructures and their fabrications by different epitaxial techniques. However, I would like to ask about several places that were not clear to me and to suggest some corrections and addings.
In Section 2
1.1. The submitted text about sample preparation looks like a preliminary version of this part. Therefore, it must be rewritten in accordance with the common rules for describing the sample fabrication (as it was perfectly described in your previous manuscripts). Some details of specific recipes can be found in supplementary, which can be added to the manuscript.
OK. We changed the corresponding text following our previous publication. Now the description of sample fabrication reads as follows:
“All the samples were grown on semi-insulating GaAs substrates by conventional solid-source molecular-beam epitaxy system (MBE32 by Riber) with the following recipes.
Sample (311)A [51]. First a 2 μm-thick Al0.55Ga0.45As layer was grown at 500â—¦C followed by a 136 nm-thick Al0.26Ga0.74As core-layer grown at 610â—¦C. At the center of the core-layer, GaAs Qds were formed by droplet epitaxy: nominally 1.5 monolayers of Ga were grown at a speed of about 0.1 monolayers per second, supplied in absence of As4 flux at 275â—¦C for the droplets formation. The droplets were then crystallized into GaAs QDs by supplying a flux of As4 (2×10−6Torr beam equivalent pressure) at 200â—¦C. The temperature was risen up to 400â—¦C for 10 minutes under As4 to improve the crystal quality of the QDs. The QDs were capped with a 30 nm-thick Al0.26Ga0.74As at 400â—¦C and the rest of the Al0.26Ga0.74As (38 nm) layer was grown at 625â—¦C. Finally, once the entire growth sequence was completed, a rapid thermal annealing process was performed at 785â—¦C for 4 minutes in an As4 atmosphere to improve the optical quality [16,52].
Sample (001) [53]. First, a thick Al0.3Ga0.7As barrier layer was grown at 580â—¦C. Then, the substrate55temperature was lowered at 350â—¦C together with reduction of the As pressure. At this point, 1.556monolayers of Ga were supplied for Ga droplet formation. Then the As4 flux was increased to 2×10−4 Torr (beam equivalent pressure) to crystallize Ga droplets into GaAs QDs at 200â—¦C that were annealed in situ at 400â—¦C for 10 minutes under As4 flux. 40 nm thick Al0.3Ga0.7As capping layer was then grown by standard MBE at 400â—¦C followed by the growth of a 20 nm thick Al0.3Ga0.7As layer at 580â—¦C. A 10 nm thick GaAs capping layer was grown by standard MBE at at 580â—¦C. Finally, the sample was processed with post-growth annealing [16,52].
Sample (111). First, a thick Al0.3Ga0.7As barrier layer was grown at 500â—¦C. Then the substrate temperature was lowered to 400â—¦C together with a reduction of the As pressure. At this point, 0.05 monolayers of Ga were supplied for Ga droplet formation. The As4 flux was set to 2×10−6 Torr beam equivalent pressure to crystallize Ga droplets into GaAs QDs at 200â—¦C that were then annealed in situ at 500â—¦C for 10 minutes under As4 flux irradiation. A 50 nm thick Al0.3Ga0.7As capping layer was grown by standard MBE at 500â—¦C followed by the growth of a 10 nm thick GaAs layer grown at 500â—¦C. Finally the sample was annealed at 600â—¦C under As4 flux to improve the optical quality [16,52]. QDs morphology was studied on samples left uncapped. We used an atomic force microscope (AFM, SPA400 by Hitachi High-Tech) in non-contact mode and an in situ STM microscope (only for the (111)A sample).
1.2. The different samples of QDs grown on (001) surface are used in the different figures of the manuscript: one is described in Fig. 2 and Fig.6, while another ones appear in Fig.3c (reference to [38]) and fig.4b (without reference). The reference to only old reference [46] should be corrected and the equality or differences of the growth processes should be compared.
All the data relative to the 001 case correspond to the same sample described in reference [Mano2009b]. In order to clarify this point we added the following comment in the captions of the corresponding figures:
“...for the sample described in reference [Mano2009b]”
For the sake of thoroughness, when relevant, we also added a similar comment for the 311A sample:
“….for the sample described in [Abbarchi2020]”
1.3. Section 2.2 describing the microscopes should be added with renumbering of optical spectroscopy to 2.3.
OK. We added a sub-section “Microscopy for morphological characterization”
In Section 3
2.1. The A, B, and C highlight cuts in different crystallographic directions would be very helpful in Figure 1a,b,c. (as it has been added in your previous publication in Nanomaterials). They can be added as insets to the figures with simultaneous displacement of the QD images.
OK. We followed the suggestion of the referee and we added the cross-sectional profiles of the QDs grown on (311)A, (100) and (111)A. Also, the figure caption is updated accordingly.
2.2 In section 3.1 a brief discussion about wetting layers will be desirable to illustrate formation/absence of this layers by TEM images. What are the main factors determining the arising of wetting layers in the QD-structures grown on the various substrates used in this work? In the previous work published in Nanomaterials, 2020, 10, 1833, the possibility of growth QDs on the surface (311)A with and without wetting layer has been reported. Did you measure binding energies and FSSs for these structures?
As these aspects have been already addressed in past publications, we refer to these papers for the TEM characterization.
The absence of a wetting layer is an advantage for high density QDs (e.g. for laser emission) and for having a larger lateral confinement. Thus, we do not have data for 311 QD with a wetting layer.
We added the following comments to clarify these points:
“The presence or absence of a wetting layer underneath DE QDs has been extensively studied in the past years (Jpn. J. Appl. Phys. 39, (2000) L79, J. Cryst. Growth 301-302 (2007) 740, Appl. Phys. Lett. 96, 062101 (2010), Appl. Phys. Lett. 98, 193112 (2011), Appl. Phys. Express 3, 065203 (2010)). Here we summarize the main phenomenology for the three surfaces.
On the GaAs(100) surface, a wetting layer is formed just before the droplet formation. Normally, we first supply Ga on the c(4ï‚´4) reconstructed surface where 1~1.75 monolayers excess As atoms are present (Phys. Rev. Lett. 92, 236105 (2004)). The first 1-1.75 monolayers of Ga atoms combine with these excess As, forming a GaAs wetting layer (Crystal Growth & Design. 14 (2014) 3110-3115). At this point Ga droplets start forming. The WL formation was clearly confirmed by cross-sectional observations (Jpn. J. Appl. Phys. 39, (2000) L79, J. Cryst. Growth 301-302 (2007) 740).
On the (311)A ((8ï‚´1) reconstruction) and (111)A surfaces ((2ï‚´2) reconstruction), the surfaces are originally Ga-rich (Phys. Rev. B 51, 14721 (1995), Phys. Rev. B 41, 3226 (1990)). On these surfaces, droplet nucleation occurs immediately after the supply of Ga (even less less than 1 mono layer of Ga (Appl. Phys. Express 3, 065203 (2010))) and the formation of a two-dimensional GaAs wetting layer is suppressed. In principle, a wetting layer can be also intentionally introduced on these surfaces by growing a GaAs layer on the AlGaAs barrier before the droplet formation (J. Crystal Growth 253 (2003) 71, J. Appl. Phys. 128, (2020) 055701).”
2.3. I recommend transferring STM images from Fig.5c to Section 3.1 and to describe them here.
OK. We moved that part in Section 3.1.
2.4. I recommend adding the title “Photon energy (meV)” for the bottom axis in fig. 2b. In addition, it would be desirable to add the bar illustrating the PL intensity plotted in this figure. Please also check the intensities of the linearly polarized components in Fig. 2a and 2b (if I understood correctly the brightness of the strokes in Fig.2b as intensity).
We believe that this information is redundant and we removed it from the corresponding figure. This characterization is well represented in the section devoted to the hh-lh mixing.
2.5. All axes in Fig.3 should be of the same thickness (slightly thicker than those used) and the symbols should be drawn with thicker lines (or filled as in Fig.4).
OK. We changed the figure according to the referee’s suggestions.
2.6. Line 157,159. Please explain what «smaller nanostructures» are? Moreover, abstract reports about size dependences. However, neither absolute nor comparative analysis of QD sizes has been performed in the manuscript. Thus, information on QD’s size distribution must be added in Section 3.1.!
OK. We added the following comment to the new version of the paper:
“Although a direct assessment of the QDs size is not possible, we can estimate the size changes based on the agreement between the data and the theoretical model shown in reference [45] as well as on magneto-optical measurements [56]. We deduce that the lateral size of the QDs for the (311)A and (111)A cases changes from about 12 nm in diameter for low energy QDs up to a minimum of about 7 nm for the high energy ones. For (001) QDs we estimate a change from about 10 nm to 4 nm”
2.7. All axes in Fig.4 should be of the same thickness (slightly thicker than those used).
OK. We changed the figure according to the referee’s suggestions.
2.8. In the caption in Fig.5 it would be desirable to add “…for an array of different (111)A QDs…” to emphasized the fact that the PL measurements were carried out for an array of the QDs. Moreover, the following text about “…the polarization analysis shown in Figure 2b” should be deleted or modified because this figure shows this analysis for QDs grown on the (001) surface.
OK. We followed the indication of the referee: the caption now reads as follows:
“a) Polarization angle q(Emax(X)) corresponding to the high energy split component of the X doublet for (111)A QDs as a function of the corresponding X emission energy obtained from a set of about 60 QDs.”
2.9. In addition, the free space in figure 5 without STM images (see p.2.2) can be used to demonstrate a similar dependence of the polarization angle on the PL peak energy measured for one of the types of asymmetric QDs grown on the (001) or (311)A surfaces. A comparative analysis of these dependences for symmetric and asymmetric QDs should be added to the text to this figure.
OK. We added to this figure also the example of the (311)A case as suggested by the referee and added the corresponding text in the body of the paper.
2.10. A legend describing the orientation of the linearly polarized PL components denoted by the red circles and black squares should be added in Figure 6. There may be a schematic sketch illustrating the orientation of the polarization and crystallographic axes, as well as the crystallographic direction of the QD elongation will be helpful for the reader in this figure.
OK. We changed the corresponding figure 7 (old figure 6), by adding a caption for the cases in a) and b) in order to show how the Qds are elongated and oriented with respect to the in-plane crystallographic axes.
2.11. Lines 237-246. The discussion on the figure proposed in p.2.9 will be helpful in this paragraph.
OK. We followed the suggestion of the referee and we added the following discussion that reads as follows:
“A similar investigation done for (311)A QDs shows that preferential directions of the polarization axes appear as expected for asymmetric structures (Figure 6 c) and d)). The case of (001) case is not reported explicitly, however, with few exceptions, all the QDs on this surface are well oriented along the [110] in plane direction, as already shown in previous works on this sample [45,84]”